# The Systemic Redox Status Is Maintained in Non-Smoking Type 2 Diabetic Subjects Without Cardiovascular Disease: Association with Elevated Triglycerides and Large VLDL

**DOI:** 10.3390/jcm9010049

**Published:** 2019-12-24

**Authors:** Peter R. van Dijk, Amaal Eman Abdulle, Marian L.C. Bulthuis, Frank G. Perton, Margery A. Connelly, Harry van Goor, Robin P.F. Dullaart

**Affiliations:** 1Department of Endocrinology, University Medical Center Groningen, University of Groningen, 9700 RB Groningen, The Netherlands; r.p.f.dullaart@umcg.nl; 2Department of Internal Medicine, division vascular medicine, University Medical Center Groningen, University of Groningen, 9700 RB Groningen, The Netherlands; 3Department of Pathology and Medical, Biology, Section Pathology, University Medical Center Groningen, University of Groningen, 9700 RB Groningen, The Netherlandsh.van.goor@umcg.nl (H.v.G.); 4Laboratory Center, University Medical Center Groningen, University of Groningen, 9700 RB Groningen, The Netherlands; F.G.Perton@umcg.nl; 5Laboratory Corporation of America® Holdings (LabCorp), Morrisville, NC 27560, USA; connem5@labcorp.com

**Keywords:** adiponectin, free thiols, nuclear magnetic resonance spectroscopy, phospholipid transfer protein activity, triglycerides, type 2 diabetes mellitus, large very low density lipoproteins

## Abstract

Decreased circulating levels of free thiols (R-SH, sulfhydryl groups) reflect enhanced oxidative stress, which plays an important role in the pathogenesis of cardiometabolic diseases. Since hyperglycemia causes oxidative stress, we questioned whether plasma free thiols are altered in patients with type 2 diabetes mellitus (T2DM) without cardiovascular disease or renal function impairment. We also determined their relationship with elevated triglycerides and very low density lipoproteins (VLDL), a central feature of diabetic dyslipidemia. Fasting plasma free thiols (colorimetric method), lipoproteins, VLDL (nuclear magnetic resonance spectrometry), free fatty acids (FFA), phospholipid transfer protein (PLTP) activity and adiponectin were measured in 79 adult non-smoking T2DM subjects (HbA1c 51 ± 8 mmol/mol, no use of insulin or lipid lowering drugs), and in 89 non-smoking subjects without T2DM. Plasma free thiols were univariately correlated with glucose (r = 0.196, *p* < 0.05), but were not decreased in T2DM subjects versus non-diabetic subjects (*p* = 0.31). Free thiols were higher in subjects with (663 ± 84 µmol/L) versus subjects without elevated triglycerides (619 ± 91 µmol/L; *p* = 0.002). Age- and sex-adjusted multivariable linear regression analysis demonstrated that plasma triglycerides were positively and independently associated with free thiols (β = 0.215, *p* = 0.004), FFA (β = 0.168, *p* = 0.029) and PLTP activity (β = 0.228, *p* = 0.002), inversely with adiponectin (β = −0.308, *p* < 0.001) but not with glucose (β = 0.052, *p* = 0.51). Notably, the positive association of free thiols with (elevated) triglycerides appeared to be particularly evident in men. Additionally, large VLDL were independently associated with free thiols (β = 0.188, *p* = 0.029). In conclusion, circulating free thiols are not decreased in this cohort of non-smoking and generally well-controlled T2DM subjects. Paradoxically, higher triglycerides and more large VLDL particles are likely associated with higher plasma levels of thiols, reflecting lower systemic oxidative stress.

## 1. Introduction

Reactive oxygen species (ROS) play essential roles in cell signalling and homeostasis [1,2]. Enhanced oxidative stress results from an imbalance between ROS production and antioxidant defence, and is prominently involved in the pathogenesis of cardiometabolic disorders including diabetes mellitus and atherosclerotic cardiovascular disease (CVD) [3]. Yet, current information with respect to the extent to which systemic oxidative stress is enhanced in the diabetic state is still limited. It has been well established that systemic ROS are increased under chronic hyperglycemic circumstances leading to progression of beta-cell deterioration, insulin resistance and atherosclerosis [4]. Remarkably however, it is not well known whether systemic oxidative stress is already enhanced in patients with type 2 diabetes mellitus (T2DM) free of clinically manifest CVD, chronic kidney disease (CKD) and albuminuria.

Thiols are organic compounds that are characterised by the presence of a sulfhydryl (R-SH) moiety [5]. R-SH groups are readily oxidised by ROS and other reactive species. The plasma concentration of free thiols has been proposed to mirror systemic or local ROS production in such a way that a decline in circulating free thiols indicates an enhanced oxidative tone [6,7]. In line with this supposition, decreased plasma levels of free thiols have been observed among persons with CVD as compared to healthy subjects [8,9]. In stable heart failure patients, higher levels of serum free thiols were associated with favourable disease outcomes [7]. In hemodialyis patients, free thiols tend to be decreased in the context of coinciding non-alcoholic fatty liver disease [10]. Furthermore, among renal transplant recipients higher free thiols concentrations were associated with a beneficial cardiovascular risk profile and a better patient and graft survival [11]. Recently, we demonstrated that in T2DM patients treated in primary care with high circulating free thiols concentrations were associated with lower HbA1c levels and less microvascular complications [12].

Elevated triglycerides, which are mainly carried by plasma very low density lipoproteins (VLDL), are a central hallmark of diabetic dyslipidemia [13,14,15]. Despite intense study, the pathogenesis of altered (hepatic) VLDL metabolism in the diabetic state has still not been fully elucidated. Among other mechanisms, enhanced free fatty acid (FFA) and glucose availability contribute to increased hepatic VLDL production in the context of T2DM [13,16,17]. Notably, the secretory control of VLDL particles by the liver is a complex process, governed by endoplasmic reticulum (ER)/proteasome-associated degradation, post ER pre-secretory proteolysis and receptor-mediated degradation, i.e., re-uptake mediated via low density lipoprotein (LDL) receptors and proteoglycan abundancy [18,19,20,21]. Of particular note, in vitro studies have revealed that intra-hepatic degradation of VLDL particles via post ER pre-secretory proteolysis is triggered by intracellular ROS, coinciding with an enhanced oxidant tone [18,21,22]. Given the role of the liver in the metabolism of fatty acids, glucose and other substrates it is not surprising that the liver is crucially involved in generating ROS [23]. Additionally, since the plasma pool of free thiols is mainly associated with albumin, it is conceivably that the liver contributes at least in part to systemic ROS status. Combined, these findings make it plausible to hypothesise that systemic ROS status is linked to triglyceride metabolism. 

The present study was initiated, firstly, to discern the extent to which plasma levels of free thiols, as a proxy of systemic ROS status, are altered in T2DM subjects without clinically manifest CVD disease and preserved renal function. Secondly, we aimed to determine associations of plasma triglycerides and VLDL particle characteristics with free thiols, thereby testing a hitherto unexplored relationship of elevated triglycerides and systemic ROS.

## 2. Experimental Section

### 2.1. Subjects and Clinical Measurements

The study was performed at the University Medical Center Groningen (Groningen, the Netherlands) and was approved by the local medical ethics committee (METC2002/174c). Caucasian participants of European descent (aged >18 years) were recruited by advertisement. Written informed consent was obtained from all participants. T2DM had been diagnosed by primary care physicians using guidelines from the Dutch College of General Practitioners (fasting plasma glucose ≥7.0 mmol/L and/or non-fasting plasma glucose ≥11.1 mmol/L). Only T2DM patients who were treated with diet alone or in combination with metformin and/or sulfonylurea were allowed to participate. T2DM subjects using insulin or other glucose lowering medication were excluded. Further exclusion criteria were clinically manifest CVD, renal insufficiency (estimated glomerular filtration rate (e-GFR) <60 mL/min/1.73 m^2^ and⁄or urinary albumin >20 mg/L). e-GFR was calculated based on the serum creatinine-based Chronic Kidney Disease Epidemiology Collaboration equation [24]. Subjects with liver disease (serum transaminase levels >2 times above the upper reference limit), pregnant women and subjects who were using lipid lowering drugs were also excluded. Subjects who used other medications (except for oral contraceptives), current smokers and subjects who used >3 alcoholic drinks daily were additionally excluded. Subjects using anti-hypertensive medication were allowed to participate. Physical examination did not reveal pulmonary or cardiac abnormalities. BMI (kg/m^2^) was calculated as weight (kg) divided by height (m) squared. Waist circumference was measured as the smallest circumference between rib cage and iliac crest. Blood pressure was measured after 15 min of rest at the left arm using a sphygmomanometer. The participants were evaluated between 08.00 and 10.00 h after an overnight fast. 

The metabolic syndrome (MetS) was defined according to NCEP-ATP III criteria [25]. Three or more of the following criteria were required for categorization of subjects with MetS: waist circumference > 102 cm for men and > 88 cm for women; hypertension (blood pressure ≥130/85 mmHg or the use of antihypertensive drugs); fasting plasma triglycerides ≥1.70 mmol/L; HDL cholesterol <1.00 mmol/L for men and <1.30 mmol/L for women; fasting glucose ≥5.60 mmol/L. Insulin sensitivity was estimated by homeostasis model assessment of insulin resistance (HOMA-IR) applying the following equation: fasting plasma insulin (mU/L) × glucose (mmol/L)/22.5 [26].

### 2.2. Laboratory Measurements

EDTA-anticoagulated plasma samples were stored at −80 °C until analysis, except for plasma glucose and glycated hemoglobin (HbA1c) levels which were measured shortly after blood collection. Glucose was measured on an APEC glucose analyser (APEC Inc., Danvers, MA, USA). 

High sensitivity C-reactive protein (hs-CRP) was measured by nephelometry with a threshold of 0.18 mg/L (BNII, Dade Behring). Glycated hemoglobin (HbA1c) was measured by high performance liquid chromatography (Bio-Rad, Veenendaal, The Netherlands). Plasma total cholesterol and triglycerides were assayed by routine enzymatic methods (Roche/Hitachi cat nos 11,876,023 and 11,875,540, respectively; Roche Diagnostics GmBH, Mannheim, Germany). HDL cholesterol was measured with a homogeneous enzymatic colorimetric test (Roche/Hitachi, cat no 04,713,214; Roche Diagnostics GmbH, Mannheim, Germany). LDL cholesterol was calculated by the Friedewald formula if plasma triglycerides were <4.5 mmol/L. Apolipoprotein B (apo B) was measured by immunoturbidimetry (Roche/Cobas Integra Tina-quant, cat. Number 03032574, respectively; Roche Diagnostics, Roche Diagnostics GmBH, Mannheim, Germany). Plasma free fatty acids (FFA) were measured with a kit from Wako Diagnostics (HR Series NEFA-HR (2), Wako Chemicals GmbH, Neuss, Germany). 

VLDL particle concentrations and VLDL subfractions were measured by nuclear magnetic resonance (NMR) spectroscopy with the LP3 algorithm (LabCorp, Morrisville, North Carolina, USA) [27,28]. VLDL particle classes and subfractions were quantified from the amplitudes of their spectroscopically distinct lipid methyl group NMR signals. The diameter range estimate for VLDL (including chylomicrons if present) was >60 nm to 29 nm, The VLDL particle concentration (VLDL-P) was calculated as the weighted average of the respective lipoprotein subclasses. The intra-assay coefficient of variation (CV) for VLDL-P amounts to 11.0%. 

Plasma PLTP activity was assayed with a phospholipid vesicles-HDL system, using (^14^C)-labelled dipalmitoyl phosphatidylcholine as described [29,30]. Briefly, plasma samples (1 μL) were incubated with (^14^C)-phosphatidylcholine-labelled phosphatidylcholine vesicles and excess pooled normal HDL for 45 min. at 37 °C. The method is specific for PLTP activity. Plasma PLTP activity levels vary linearly with the amount of plasma added to the incubation system. PLTP activity was related to the activity in human reference pool plasma and was expressed in arbitrary units (AU; 100 AU corresponds to 13.6 μmol phosphatidylcholine transferred per mL per h). The intra-assay CV of PLTP activity was 5%. 

Insulin was assayed by microparticle enzyme immunoassay (AxSYM insulin assay: Abbott Laboratories, Abbott Park, IL, USA). Total adiponectin was measured by enzyme-linked immunosorbent assay (Linco Research Inc., St. Charles, MO, USA, cat no EZHADP-61k). The provided by this assay are strongly correlated with results obtained by enzyme-linked immunoassay measurement [31]. 

Plasma free thiols were assayed as described with minor modifications [32,33]. After thawing, samples were four-fold diluted using 0.1 M Tris buffer (pH 8.2). Background absorption was measured at 412 nm using the Varioskan microplate reader (ThermoScientific, Breda, the Netherlands), together with a reference measurement at 630 nm. Subsequently, 20 μL 1.9 mM 5,5′-dithio-bis (2-nitrobenzoic acid) (DTNB, Ellman’s Reagent, CAS no. 69-78-3, Sigma Aldrich Corporation, St. Louis, MO, USA) was added to the samples in 0.1 M phosphate buffer (pH 7.0). After an incubation time of 20 min at room temperature, absorbance was measured again. Final serum free thiol concentrations were established by parallel measurement of an L-cysteine (CAS no. 52-90-4, Fluka Biochemika, Buchs, Switzerland) calibration curve (range from 15.6 μM to 1000 μM) in 0.1 M Tris/10 mM EDTA (pH 8.2). The intra-assay CV of free thiols amount to 6%. 

### 2.3. Statistical Analysis 

SPSS (IBM SPSS Statistics for Windows, Version 23.0. IBM Corp. Armonk, NY, USA) was used for data analysis. The distributions of all variables were examined using histograms and Q-Q plots. Data are expressed as mean (with standard deviation (SD)) or median (with interquartile range (IQR)) for normally distributed and non-normally distributed data, respectively. Nominal data are presented as n (with percentage (%)). Non-parametrically distributed data were log_e_ transformed for statistical analysis. Between-group differences in continuous variables were determined by unpaired T-tests. Between-group differences in dichotomous variables were determined by χ^2^-analysis. Univariate relationships were assessed using Pearson correlation coefficients. Multivariable linear regression analysis was applied to disclose the independent contribution of variables. A *p*-value <0.05 (two-sided) was used to indicate statistical significance. 

## 3. Results

### 3.1. Study Population

The study population comprised 79 T2DM subjects and 89 non-diabetic subjects (Table 1). Median diabetes duration was 5 (interquartile range (IQR) 4–6.5) years. Fourteen T2DM subjects used metformin and 11 used sulfonylurea alone, whereas these drugs were used combined in 24 patients. No other glucose lowering drugs were taken. The other T2DM subjects were given lifestyle advice only. Anti-hypertensive medication (mostly angiotensin-converting enzyme inhibitors, angiotensin II receptor antagonists and diuretics, alone or in combination) were taken by 33 T2DM subjects and by none of the non-diabetic subjects (*p* < 0.001). Estrogens were used by two pre- and two postmenopausal non-diabetic women. Other medications were not used.

As shown in Table 1, T2DM subjects were on average 4 years older than the non-diabetic subjects, were more obese, had higher blood pressure, fasting plasma glucose and HbA1c levels, were more insulin resistant and categorised with MetS more frequently. Sex distribution was not different between the groups. e-GFR was similar in the groups. 

### 3.2. Comparison of Laboratory Variables between Subject Categories 

Plasma total cholesterol was lower in T2DM subjects, which was due to lower HDL cholesterol. Triglycerides were higher in T2DM subjects but non-HDL cholesterol, LDL cholesterol and apolipoprotein b (ApoB) levels were not different between the groups. The total VLDL particle concentration was not different between the groups, but large VLDL particles were increased, whereas small VLDL particles tended to be decreased in T2DM subjects. Plasma FFA and PLTP activity were elevated, whereas adiponectin was decreased in T2DM subjects (Table 1). 

In all subjects combined, plasma free thiol concentrations were higher in men (n = 98, 655 ± 87 µmol/L) than in women (n = 70, 614 ± 91 µmol/L, *p* = 0.003). Free thiols were not different between T2DM and non-diabetic subjects (Table 1), and were not significantly related to diabetes duration (r = −0.136, *p* = 0.23). Free thiols were also not different in T2DM subjects compared to non-diabetic subjects after exclusion of T2DM subjects using glucose lowering medication (n = 20, 600 ± 109 µmol/L, *p =* 0.14), antihypertensive medication (n = 44, 652 ± 110 µmol/L, *p* = 0.21) or both classes of drugs (n = 15, 598 ± 117µmol/L, *p =* 0.17). Likewise, free thiols were also not different between subjects with MetS (n = 75, 643 ± 89 µmol/L) and without MetS (n = 93, 633 ± 92 µmol/L, *p* = 0.49). Free thiols were not different in subjects with elevated glucose (*p* = 0.39), enlarged waist circumference (*p* = 0.44) or elevated blood pressure (*p* = 0.29) compared to subjects who did not fulfil these MetS criteria (data not shown). Notably, free thiols were higher in subjects with elevated triglycerides (n = 71, 663 ± 84 µmol/L, *p* = 0.002) or with low HDL cholesterol (n = 45, 661 ± 84 µmol/L, *p* = 0.042) compared to subjects who did not fulfil these MetS criteria (triglyceride criterion: 619 ± 91 µmol/L; HDL cholesterol criterion (629 ± 92 µmol/L).

### 3.3. Univariate Correlations

In all subjects combined, univariate analysis showed that plasma free thiols were inversely correlated with age, HDL cholesterol, FFA and adiponectin, and positively with glucose, e-GFR, non-HDL cholesterol, triglycerides, VLDL particle concentration and large-and medium-sized VLDL particles (Table 2, panel A). Comparable correlations were observed in T2DM subjects separately (Table 2, panel B and panel C). The univariate correlations of plasma free thiols with triglycerides and with large VLDL particles are shown in Figure 1, panel A and panel B, respectively. In non-diabetic subjects, free thiols were correlated positively with total cholesterol, non-HDL cholesterol, triglycerides and ApoB, and inversely with FFA and adiponectin (Table 2, panel C). Neither in all subjects combined nor in T2DM subjects and non-diabetic subjects separately were free thiols correlated with blood pressure, BMI, waist circumference and PLTP activity (Table 2, panel A, B and C). In addition, in all subjects combined, PLTP activity was correlated positively with triglycerides (r = 0.325, *p* < 0.001) and FFA (r = 0.162, *p* = 0.036; data not shown). Similar correlations of PLTP activity with triglycerides were found in T2DM subjects and non-diabetic subjects separately (data not shown). 

In all men combined (n = 98), plasma free thiols were inversely correlated with age and adiponectin, and positively with HOMA-IR, e-GFR, triglycerides, VLDL particle concentration and large VLDL particles (Appendix A). In all women combined (n = 70), plasma free thiols were not significantly correlated with any of the variables listed in Table 2 (Appendix A).

### 3.4. Multivariable Linear Regression Analyses

Multivariable linear regression analyses were first carried out to disclose the association of free thiols with the diabetic state when taking account of age and sex. In this analysis, free thiols were not significantly related to the presence of T2DM after adjustment for age and sex (*p* = 0.159; data not shown). Next, we examined the extent to which plasma free thiol concentrations were independently associated with the individual MetS components. This analysis showed that free thiols were positively associated with elevated triglycerides but not with other MetS components (Table 3). Furthermore, free thiols were positively associated with male sex (Table 3). In an additional analysis, now also including e-GFR, there was no independent association of free thiols with e-GFR (β = 0.122, *p* = 0.20), while the relationship with elevated triglycerides remained significant (β = 0.209, *p* = 0.011). In a sex-stratified analysis, plasma free thiols were positively associated with elevated triglycerides in men (Appendix A). In women, plasma free thiols were not significantly related to any of the individual MetS components (Appendix A).

Given the association between triglycerides and free thiols, we next determined whether this relationship remained when taking account of the diabetic state instead of the MetS criterion of elevated plasma glucose. In this analysis, plasma triglycerides were positively associated with free thiols (β = 0.271, *p* < 0.001) after adjustment for age (β = 0.01, *p* = 0.89), sex (β = 0.001, *p* = 0.97) and the presence of diabetes (β = 0.245, *p* = 0.059). Subsequently, we tested whether other variables that were univariately related to plasma triglycerides or free thiols including PLTP activity, FFA and adiponectin (Table 2). Plasma glucose was also included in these models in view of its univariate correlation with free thiols. In this multivariable model, with plasma triglycerides as dependent variable, plasma triglycerides were positively and independently associated with free thiols, FFA and PLTP activity, and inversely with adiponectin but not with glucose (Table 4). These associations remained significant after further adjustment for the use of sulfonylurea, metformin and antihypertensive drugs (data not shown). Similar associations of plasma triglycerides with free thiols, FFA, PLTP activity and adiponectin were found with HbA1c or the presence of T2DM instead of plasma glucose (data not shown).

In sex-stratified analysis, it was found that plasma triglycerides were positively associated with free thiols, FFA, PLTP activity and inversely with adiponectin in men (Appendix A). In women, plasma triglycerides were positively associated with PLTP activity and tended to be inversely associated with adiponectin, but the association with free thiols was not significant (Appendix A).

Finally, we determined the association of free thiols with VLDL subfractions. Large-sized VLDL particles were positively and independently associated with free thiols and FFA, and inversely with adiponectin (Table 5, panel A). Medium-sized VLDL particles tended to be associated positively with free thiols and were inversely associated with adiponectin (Table 5, panel B). Small-sized VLDL particles were not independently associated with any of these variables (Table 5, panel C). In sex-stratified analyses, the independent association of large VLDL particles with plasma free thiols did not reach statistical significance in men (β = 0.184, *p* = 0.14) or in women (β = 0.188, *p* = 0.14; data not shown).

## 4. Discussion

In this cohort of non-smoking and in general appropriately controlled T2DM subjects, we report that circulating levels of free thiols are still maintained. In the current study, we excluded T2DM subjects who smoked, those with clinically manifest CVD or impaired renal function as well as subjects using insulin or lipid lowering medication. Therefore, the present findings suggest that decreased free thiol concentrations, as a proxy of enhanced systemic oxidative stress, are unlikely to represent an early feature of relatively moderate chronic hyperglycemia. Second, we report here for the first time that plasma triglycerides and in particular large VLDL particles are positively and independently related to free thiols in multivariable linear regression analysis in which we accounted for age, sex, glucose (or alternatively the presence of diabetes), FFA, PLTP activity and adiponectin. Notably, the association of free thiols with triglycerides appeared to be particularly evident in men. We consider this novel and seemingly surprising finding to agree with the hypothesis that systemic ROS status may impact on triglyceride metabolism. As a mechanism, we postulate that this association may at least in part be attributable to an effect of hepatic ROS to inhibit VLDL release in the bloodstream.

To underscore the lack of effect of the presence of T2DM on plasma free thiol concentrations as observed in the present study, we also carried out sensitivity analyses after exclusion of T2DM subjects using any glucose lowering medication, antihypertensive medication or both classes of drugs. Again, it was found that plasma free thiols were not decreased in the cohort of T2DM subjects studied. Our results can be compared with another study on thiols in T2DM by Pasaoglu et al. [34]. In a small study among 40 T2DM subjects (20 newly diagnosed and 20 treated using oral blood glucose lowering therapy only) and 20 non-diabetic subjects they demonstrated lower plasma total thiol levels among persons with T2DM as compared to control subjects (605 ± 36 versus 665 ± 60 μmol/L) [34]. Since a negative correlation between HbA1c and thiols has been reported [12], it should be taken into account that T2DM subjects in their study were poorly regulated with an average HbA1c of 74 ± 30 mmol/mol and of 78 mmol/mol, in newly diagnosed and medically treated T2DM subjects, respectively. Recently, we measured free thiols in a population of 929 T2DM outpatients treated in primary care [12]. In this cohort, we found significant multivariable associations between free thiols and the presence of macrovascular complications, age, BMI, diastolic blood pressure, HbA1c (all inverse associations), male sex, diabetes duration and use of platelet aggregation inhibitors (all positive associations). In addition, there was (modest) additional value for free thiols for risk prediction of long-term complications. In agreement with these and previous results, we observed in the current study that free thiols declined with ageing and were higher in men than in women [11,12], whereas we also confirmed a positive relationship with e-GFR at least in univariate regression analysis [11].

Data concerning the relationship of ROS status and circulating free thiols in particular on fasting plasma triglycerides in humans is scarce. In the aforementioned study amongst primary care treated T2DM patients, there was no association between plasma free thiols and triglyceride concentrations in multivariable analysis [12]. It should be noted, however, that in this population, samples were non-fasting and 80% of the patients used cholesterol-lowering medication. Amongst persons with type 1 diabetes, no association between plasma free thiols and triglycerides was present [35]. Interestingly, in mice receiving a high-saturated fat diet, the intake of the anti-oxidative ‘thiol donor’ n-acetyl cysteine decreased plasma triglyceride concentrations possibly by lowering the activity and mRNA of malic enzyme, fatty acid synthase and 3-hydroxy-3-methylglutaryl coenzyme A reductase [36,37].

In the present study, we assessed the relationship of plasma free thiols with triglycerides not only when taking account of FFA and glucose, which are well-known substrates of hepatic triglyceride synthesis [13,16], but also of PLTP activity and adiponectin. We found that plasma triglycerides as well as large VLDL particles were associated with FFA levels in agreement with the contention that FFA provide an important source for hepatic VLDL synthesis [13,17,38]. Although hepatic overproduction of large VLDL particles, a central feature of diabetic dyslipidemia, has been proposed to be to an important extent driven by hyperglycemia [16], plasma triglycerides were not independently associated with glucose, which we ascribe to the modestly elevated fasting glucose levels in the current T2DM cohort.

Importantly, studies in mice with genetic *Pltp* deficiency have revealed that PLTP is able to promote the hepatic secretion of apoB-containing lipoproteins [39], effect which was subsequently found to be attributable to the ability of PLTP to regulate ROS-dependent intracellular degradation of newly assembled VLDL [40,41]. In humans, plasma PLTP activity has repeatedly been shown to be associated with plasma triglyceride metabolism [42,43,44,45]. Moreover, genetic variation in *Pltp* associates with lipid traits including triglycerides [46]. Plasma triglycerides were indeed positively associated with PLTP activity in the current study. Of note, this relationship was independent from its positive association with plasma free thiols. In establishing the association of plasma triglycerides with free thiols, we also accounted for plasma adiponectin in multivariable regression analysis. Adiponectin may stimulate hepatic fatty acid oxidation and ameliorate the development of hepatic steatosis [47]. In line, we previously reported an inverse relationship of plasma adiponectin with alanine aminotransferase [48]. Importantly, adiponectin has been implicated in preventing mitochondrial dysfunction, and hence, may inhibit ROS production in vitro, possibly by promoting uncoupling protein 2 expression [47]. Of note, we observed an inverse univariate relationship of plasma free thiols with adiponectin, thereby providing a rationale to adjust for plasma levels of this adipokine when evaluating the association of plasma triglycerides with free thiols. Still, plasma triglycerides as well as large VLDL particles remained associated with plasma free thiols when considering adiponectin in multivariable linear regression analysis. Taken together, our current study provides support for the hypothesis that ROS status may influence triglyceride metabolism even independent of plasma glucose, FFA, PLTP activity and adiponectin. However, since plasma free thiols were not different between the currently studied T2DM and non-diabetic subjects, it seems unlikely that alterations in ROS status explain to a considerable extent why plasma triglycerides and large VLDL particle concentrations are increased in T2DM. Additionally, the mechanisms responsible for apparent sex-differences in the association of free thiols with triglycerides are not yet clear. Plasma free thiols are higher in men than in women as confirmed in the current study [11,12]. Additionally, fasting plasma triglycerides may also be higher in men (as derived from [49]). Among other explanations it seems, therefore, plausible that there is less between subject variation in free thiols and triglycerides in women compared to men, which could to some extent blunt the strength of the correlation between free thiols and triglycerides in women.

Our findings are relevant from a diagnostic perspective. In transplant patients and heart failure patients the systemic redox status may have predictive power for outcome [7,11]. In recent work we also showed that the systemic level of free thiols is a better reflection of disease activity in inflammatory bowel disease than faecal calprotectin, the classical marker [50]. However, free thiols are also receptive to therapeutic modulation, and can therefore be considered a potential therapeutic target for therapy [51,52]. In clinical trials, a cysteine derivative N-Acetylcysteine directly reduces disulfide bonds and act thereby as a glutathione (GSH) precursor [52]. Although promising, therapeutic results are inconsistent and these compounds may be effective in selected patients with low free thiol concentrations.

Several other methodological aspects of our study need to be discussed. First, we carried out a cross-sectional study. For this reason, cause-effect relationships cannot be established with certainty, nor can we exclude the possibility of reversed causation. Hence, we cannot exclude that alterations in triglyceride metabolism could affect the balance between ROS production and antioxidant defence. In this vein, a scenario is also possible whereby increased hepatic substrate availability leads to enhanced β-oxidation which in turn affects hepatic ROS production. Second, the sample size of the current study was rather limited, possibly masking relatively modest associations between variables of interest. Third, only diabetic subjects who did not use lipid lowering medication were allowed to participate, thereby avoiding confounding due to effects of statins on triglycerides and PLTP activity [53,54]. As a consequence, it is likely that T2DM subjects with mild dyslipidemia were preferentially included. Fourth, only non-smokers participated. This exclusion criterion was implemented in order to avoid effects of smoking on systemic ROS [55]. Moreover, cigarette smoking affects lipid metabolism, including elevations in PLTP activity [56].

## 5. Conclusions

Plasma free thiols, which mirror the redox status, are still maintained in generally adequately controlled non-smoking T2DM subjects, without clinically manifest cardiovascular disease or chronic kidney disease indicating limited organ and cell damage. Higher fasting plasma triglycerides and increased large VLDL particles concentrations are inversely associated with free thiols, raising the possibility that enhanced oxidative stress could be paradoxically implicated in less severe dyslipidemia.

## Figures and Tables

**Figure 1 jcm-09-00049-f001:**
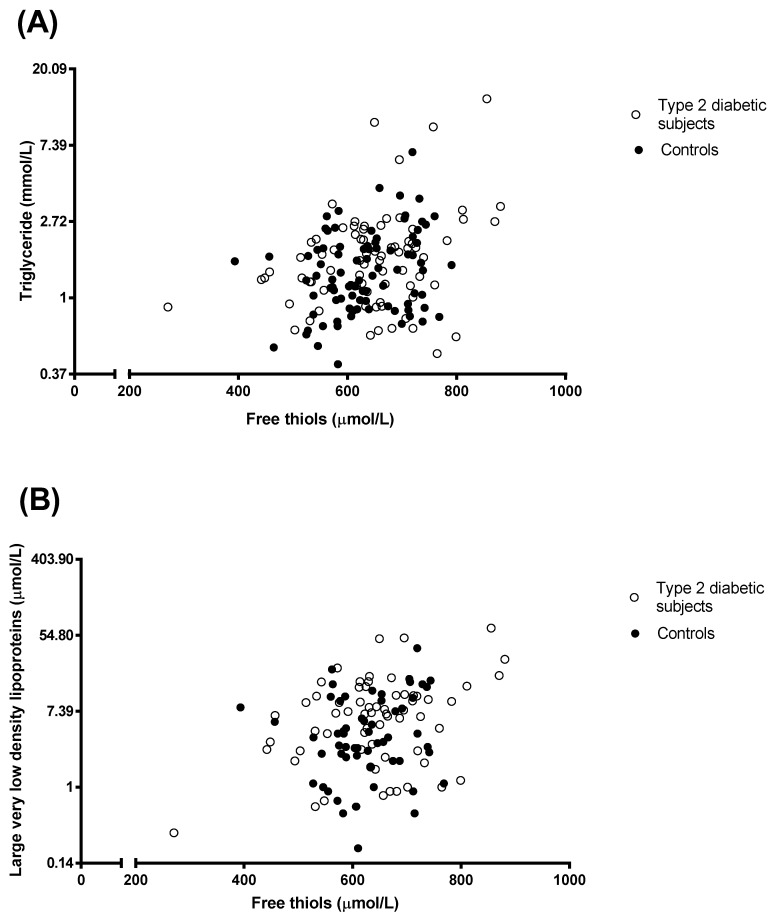
Univariate correlations of plasma free thiols with plasma triglycerides (panel **A**; 79 type 2 diabetic subjects and 89 subjects without diabetes) and with very large very low density lipoprotein (VLDL) (panel **B**; 66 type 2 diabetic subjects and 58 subjects without diabetes). Legend: closed symbols represent subjects without diabetes; open symbols represent subjects with type 2 diabetes. The log transformed data (i.e., triglyceride and large very low density lipoproteins) were used. The back transformed measures are shown on the *Y*-axis. Pearson correlation coefficients: free thiols with triglycerides; all subjects, r = 0.285, *p* < 0.001; Type 2 diabetic subjects, r = 0.290, *p* = 0.01; subjects without diabetes, r = 0.260, *p* = 0.014. Free thiols with large VLDL: all subjects, r = 0.266, *p* = 0.003; Type 2 diabetic subjects, r = 0.333, *p* = 0.006; subjects without diabetes, r = 0.138, *p* = 0.31.

**Table 1 jcm-09-00049-t001:** Clinical and laboratory characteristics in 79 subjects with Type 2 diabetes mellitus (T2DM) and 89 subjects without T2DM.

	T2DM Subjects (n = 79)	Non-Diabetic Subjects (n = 89)	*p*-Value
Age (years)	59 ± 9	55 ± 9	0.011
Sex (men/women)	49/30	49/40	0.45
Metabolic syndrome (yes/no)	54/25	21/68	<0.001
Systolic blood pressure (mm Hg)	144 ± 20	131 ± 19	<0.001
Diastolic blood pressure (mm Hg)	87 ± 9	83 ± 11	0.007
BMI (kg/m^2^)	28.6 ± 4.8	25.9 ± 3.9	<0.001
Waist (cm)	100 ±13	89 ± 9	<0.001
Glucose (mmol/L)	8.9 ± 2.3	5.6 ± 0.7	<0.001
HbA1c (mmol/mol)	51 ± 8	39 ± 5	<0.001
Insulin (mU/L)	10.8 (7.2,15.3)	6.5 (4.7,8.6)	<0.001
HOMA-IR (mU mmol/L^2^/22.5)	3.96 (2.37,6.64)	1.57 (1.09,2.29)	<0.001
e-GFR (mL./min/1.73 m^2^)	91 ± 13	87 ± 14	0.12
Total cholesterol (mmol/L)	5.38 ± 0.96	5.74 ± 0.99	0.022
Non-HDL cholesterol (mmol/L)	4.12 ± 1.04	4.25 ± 1.06	0.44
LDL cholesterol (mmol/L)	3.27 ± 0.85	3.53 ± 0.89	0.06
HDL cholesterol (mmol/L)	1.26 ± 0.38	1.49 ± 0.40	<0.001
Triglycerides (mmol/L)	1.70 (1.17,2.30)	1.31 (0.89,1.95)	0.034
ApoB (g/L)	0.95 ± 0.23	0.96 ± 0.23	0.55
VLDL-P (nmol/L)	69.5 (46.2,88.7)	61.8 (51.6,101.6)	0.83
Large VLDL (nmol/L)	7.1 (2.7,11.1)	3.5 (2.0,9.4)	0.019
Medium VLDL (nmol/L)	28.9 (14.9,41.3)	25.3 (13.2,44.8)	0.86
Small VLDL (nmol/L)	27.7 (16.8,43.9)	33.9 (21.2,44.5)	0.076
FFA (mmol/L))	0.35 ± 0.12	0.30 ± 0.12	0.001
PLTP activity (AU)	104 ± 11	94 ± 10	<0.001
Adiponectin (mg/L)	14.3 (10.7,26.3)	20.1 (14.9,43.0)	<0.001
Free Thiols (µmol/L)	645 ± 102	630 ± 78	0.31

Data are expressed in means ± SD, medians (interquartile range) or numbers. Abbreviations: ApoB: apolipoprotein b; BMI: body mass index; e-GFR: estimated glomerular filtration rate; FFA: free fatty acids; HbA1c: glycated hemoglobin; HDL: high density lipoproteins; LDL: low density lipoproteins; PLTP: phospholipid transfer protein; VLDL: very low density lipoproteins: VLDL-P: very low density particle concentration. Differences between subjects with and without T2DM were determined by T-tests (using log_e_ transformed values in case of not-normally distributed data) and by χ^2^ tests, where appropriate. VLDL particle concentrations and VLDL subfractions were measured in 66 subjects with diabetes and in 56 subjects without diabetes. LDL cholesterol was calculated by the Friedewald formula in 75 T2DM subjects and in 87 non-diabetic subjects.

**Table 2 jcm-09-00049-t002:** Univariate relationships of plasma free thiols with clinical and laboratory variables in all subjects combined (n = 168, panel **A**), Type 2 diabetic (T2DM) subjects (n = 79, panel **B**) and in non-diabetic subjects (n = 89, panel **C**).

	(A) All Subjects (n = 168) Free Thiols	(B) T2DM Subjects (n = 79) Free Thiols	(C) Non-Diabetic Subjects (n = 89) Free Thiols
Age	−0.183 *	−0.297 **	−0.098
Systolic blood pressure	−0.053	−0.074	−0.093
Diastolic blood pressure	0.031	0.145	−0.119
BMI	0.059	−0.012	0.108
Waist	0.131	0.057	0.172
Glucose	0.196 *	0.262 *	0.012
HbA1c	0.151	0.186	−0.013
Insulin	0.095	0.022	0.131
HOMA-IR	0.141	0.106	0.132
e-GFR	0.213 **	0.271 *	0.144
Total cholesterol	0.084	−0.057	0.281 **
Non-HDL cholesterol	0.168 *	0.056	0.314 **
LDL cholesterol	0.034	−0.148	0.247 *
HDL cholesterol	−0.232 **	−0.302 **	−0.137
Triglycerides	0.285 ***	0.290 **	0.260 *
ApoB	0.129	0.063	0.349 ***
VLDL-P	0.204 *	0.247 *	0.138
Large VLDL	0.266 **	0.333 **	0.129
Medium VLDL	0.234 **	0.333 **	0.081
Small VLDL	−0.013	0.007	−0.020
FFA	−0.187 *	−0.210	−0.224 *
PLTP activity	0.034	−0.096	0.133
Adiponectin	−0.261 ***	−0.259 *	−0.247 *

Pearson correlation coefficients are shown. Non-parametrically distributed data are log_e_ transformed. * *p* < 0.05; ** *p* ≤ 0.01: *** *p* ≤ 0.001. Abbreviations: ApoB: apolipoprotein B; BMI: body mass index; e-GFR: estimated glomerular filtration rate; FFA: free fatty acids: HbA1c: glycated hemoglobin; HDL: high density lipoproteins; LDL: low density lipoproteins; PLTP: phospholipid transfer protein; VLDL: very low density lipoproteins; VLDL-P: very low density particle concentration. VLDL particle concentrations and VLDL subfractions were measured in 66 subjects with diabetes and in 56 subjects without diabetes. LDL cholesterol was calculated by the Friedewald formula in 75 T2DM subjects and in 87 non-diabetic subjects.

**Table 3 jcm-09-00049-t003:** Multivariable linear regression analysis demonstrating independent associations of plasma free thiols with age, sex and individual metabolic syndrome components in 168 subjects (79 Type 2 diabetic subjects and 89 non-diabetic subjects).

	β	*p*-Value
Age	−0.205	0.011
Sex (men versus women)	0.213	0.005
Elevated glucose	0.074	0.35
Elevated blood pressure	−0.069	0.40
Enlarged waist	0.021	0.80
Elevated triglycerides	0.200	0.015
Low HDL cholesterol	0.060	0.47

Abbreviations: β: standardised regression coefficient; HDL: high density lipoproteins.

**Table 4 jcm-09-00049-t004:** Multivariable linear regression analysis demonstrating independent associations of plasma triglycerides with age, sex, free thiols, free fatty acids, phospholipid transfer protein activity and adiponectin and glucose in 168 subjects (79 Type 2 diabetic subjects and 89 non-diabetic subjects).

	β	*p*-Value
Age	−0.044	0.55
Sex (men versus women)	0.016	0.84
Free thiols	0.215	0.004
FFA	0.168	0.029
PLTP activity	0.228	0.002
Adiponectin	−0.308	<0.001
Glucose	0.052	0.51

Triglycerides and adiponectin are log_e_ transformed. Abbreviations: β: standardised regression coefficient; FFA: free fatty acids: PLTP phospholipid transfer protein.

**Table 5 jcm-09-00049-t005:** Multivariable linear regression analysis demonstrating independent associations of very low density lipoprotein (VLDL) subfractions with age, sex, free thiols, free fatty acids, PLTP activity, adiponectin and glucose in 122 subjects (66 Type 2 diabetic subjects and 56 non-diabetic subjects). Panel **A**, Large VLDL; panel **B**, medium VLDL, panel **C**, small VLDL.

	A Large VLDL		B Medium VLDL		C Small VLDL	
	β	*p*-Value	β	*p*-Value	β	*p*-Value
Age	−0.042	0.630	0.036	0.71	0.021	0.82
Sex (men versus women)	0.074	0.440	0.049	0.65	−0.156	0.17
Free thiols	0.188	0.029	0.176	0.067	−0.023	0.82
FFA	0.242	0.008	−0.014	0.91	−0.157	0.41
PLTP activity	0.147	0.094	0.075	0.45	−0.066	0.52
Adiponectin	−0.294	0.002	−0.243	0.021	−0.047	0.67
Glucose	0.097	0.320	−0.052	0.630	0.029	0.80

VLDL subfractions and adiponectin are log_e_ transformed. Abbreviations: β: standardised regression coefficient; FFA: free fatty acids; PLTP phospholipid transfer protein; VLDL: very low density lipoproteins.

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
