# Peer review of "The Systemic Redox Status Is Maintained in Non-Smoking Type 2 Diabetic Subjects Without Cardiovascular Disease: Association with Elevated Triglycerides and Large VLDL"

_jcm, 2019, doi:10.3390/jcm9010049_

Round 1

Reviewer 1 Report

Title: The Systemic Redox Status is Maintained in Non-Smoking Type 2 Diabetic Subjects without Cardiovascular Disease: Association with Elevated
Triglycerides and Large VLDL

Suggestions:

Association of fasting plasma triglycerides and large VLDL particles with free thiols is definitely and interesting finding. Do you think its possible to show a  XY scatter plot in a figure?

Experimental section will be more clear with few subsections. Result section will be more clear with subsections. Table 2 has A,B and C. Table 2 has been mentioned as Table 2 only in results, please mention A,B and C also. Also put brackets to (A),(B) and (C) to show them more clearly.

Table 4 A written in results can be changed to Figure 4. As there is no A. Please mention Figure 5 A(large VLDL),B(Medium VLDL) and C(Small VLDL) and mention more clearly A,B, and C. Please check your punctuation.

Example: Line 116, Line 339,Line 369. Line 312 in change the font ..its bold [42-45],while rest of the references are in normal font.

Author Response

Reply to the reviewers

Title:

The Systemic Redox Status is Maintained in Non-Smoking Type 2 Diabetic Subjects without Cardiovascular Disease: Association with Elevated Triglycerides and Large VLDL

Authors:

Peter R. van Dijk, Amaal Eman Abdulle, Marian L.C. Bulthuis, Frank G. Perton, Margery A. Connelly, Harry van Goor, Robin P.F. Dullaart

Manuscript ID:

jcm-645567

P.R. van Dijk

University of Groningen, University Medical Centre, Dept. of Internal Medicine.

Hanzeplein 1, 9700 RB, HP AA41, Groningen, The Netherlands.

Reviewer 1

Association of fasting plasma triglycerides and large VLDL particles with free thiols is definitely and interesting finding. Do you think its possible to show a  XY scatter plot in a figure?

We added figures that demonstrate the correlation of free plasma thiols with triglycerides and very large very low density lipoprotein.

Experimental section will be more clear with few subsections. Result section will be more clear with subsections. Table 2 has A,B and C. Table 2 has been mentioned as Table 2 only in results, please mention A,B and C also. Also put brackets to (A), (B) and (C) to show them more clearly.

According to the reviewers’ suggestion we added subsections to the experimental and results section and mentioned the different parts (A, B and/or C) in the text.

Table 4 A written in results can be changed to Figure 4. As there is no A. Please mention Figure 5 A(large VLDL),B(Medium VLDL) and C(Small VLDL) and mention more clearly A,B, and C. Please check your punctuation.

Example: Line 116, Line 339,Line 369. Line 312 in change the font ..its bold [42-45],while rest of the references are in normal font.

We changed this part of the results section and improved punctuation throughout the paper.

Reviewer 2 Report

Suggested revisions.

The study is interesting and give important information for understanding associations between oxidative stress and disturbed metabolic pattern, not only with T2DM but also with obesity related disturbance (metabolic syndrome). However, it is a rather small population that grasp over many important variables. The major comment concerns the analysis on the total subjects with a multivariable regression analysis, age and sex adjusted. The strong association between sex and thiol in table 3 indicate need for stratification.

I suggest an additional multiple regression analysis for males and females separately (split for sex) to assess associations between both thiol and triglycerides with all variables included in table 3 and 4. This analysis should be performed on all subjects (n=168) with adjustments for T2DM. I suggest that the univariate associations between thiol and other variables need also to be presented for males and females separately. I suggest the discussion to be improved with added comments and focus on T2DM and sex differences.

Minor comments. Decide to use either gender or sex in the manuscript (males, females indicate sex)

Author Response

Reply to the reviewers

Title:

The Systemic Redox Status is Maintained in Non-Smoking Type 2 Diabetic Subjects without Cardiovascular Disease: Association with Elevated Triglycerides and Large VLDL

Authors:

Peter R. van Dijk, Amaal Eman Abdulle, Marian L.C. Bulthuis, Frank G. Perton, Margery A. Connelly, Harry van Goor, Robin P.F. Dullaart

Manuscript ID:

jcm-645567

P.R. van Dijk

University of Groningen, University Medical Centre, Dept. of Internal Medicine.

Hanzeplein 1, 9700 RB, HP AA41, Groningen, The Netherlands.

Reviewer 2

The study is interesting and give important information for understanding associations between oxidative stress and disturbed metabolic pattern, not only with T2DM but also with obesity related disturbance (metabolic syndrome). However, it is a rather small population that grasp over many important variables. The major comment concerns the analysis on the total subjects with a multivariable regression analysis, age and sex adjusted. The strong association between sex and thiol in table 3 indicate need for stratification.

I suggest an additional multiple regression analysis for males and females separately (split for sex) to assess associations between both thiol and triglycerides with all variables included in table 3 and 4. This analysis should be performed on all subjects (n=168) with adjustments for T2DM. I suggest that the univariate associations between thiol and other variables need also to be presented for males and females separately. I suggest the discussion to be improved with added comments and focus on T2DM and sex differences.

According to the reviewers’ suggestion we added these analyses as supplemental material to our paper. The association of free thiols with triglycerides appeared to be most evident in men. In the result and discussion section of our paper we discussed these outcomes.

Minor comments. Decide to use either gender or sex in the manuscript (males, females indicate sex)

We now use ‘sex’ throughout the paper instead of ‘gender’ throughout our paper.

Round 2

Reviewer 2 Report

Thank you for adding analyses on sex differences and discussion according to this.